# Preconditioned Crank-Nicolson Algorithms for Wide Bayesian Neural Networks

**Lucia Pezzetti**[*]
ETH AI Center
lucia.pezzetti@ai.ethz.ch

**Stefano Favaro**
University of Torino
and Collegio Carlo Alberto
stefano.favaro@unito.it

**Stefano Peluchetti**
Cogent Labs
speluchetti@cogent.co.jp

## Abstract

Bayesian Neural Networks represent a fascinating confluence of deep learning techniques and probabilistic reasoning, offering a compelling framework for understanding uncertainty in complex predictive models. In this paper, we consider Bayesian Neural Networks with Gaussian initialization and we investigate the use of the preconditioned Crank-Nicolson algorithm to sample from the reparametrized posterior distribution of the weights as the width of the network grows. In addition to being robust in the infinite-dimensional setting, we prove that the acceptance probability of the preconditioned Crank-Nicolson sampler approaches 1 as the width of the network goes to infinity, independently of any stepsize tuning. We then compare how the efficiency of the Langevin Monte Carlo, the preconditioned Crank-Nicolson and the preconditioned Crank-Nicolson Langevin samplers are influenced by changes in the network width in some real-world cases. In particular, we demonstrate that in wide Bayesian Neural Networks configurations, the proposed method allows for more efficient sampling, as evidenced by a higher effective sample size and improved diagnostic results compared with the Langevin Monte Carlo algorithm.

## 1 Introduction

Bayesian Neural Networks (BNNs) have emerged as a powerful framework for combining deep learning with probabilistic reasoning, offering a principled approach to understanding uncertainty in complex predictive models (1; 2; 3; 4; 5; 6; 7). Despite their potential advantages, BNNs face significant challenges, particularly in sampling from high-dimensional posterior distributions of network weights. As the width of neural networks increases, standard Markov Chain Monte Carlo (MCMC) methods often struggle with efficiency and scalability. In this paper, we give theoretical and empirical guarantees to address these challenges by leveraging function-space MCMC techniques, specifically the preconditioned Crank-Nicolson (pCN) algorithm (8), for sampling from the posterior distributions of wide BNNs. Our method exploits recent theoretical advances in the understanding of wide neural networks and offers a robust sampling procedure that remains effective as network width increases.

---

[*]Work done when Lucia Pezzetti was at the University of Torino

Workshop on Bayesian Decision-making and Uncertainty, 38th Conference on Neural Information Processing Systems (NeurIPS 2024).

## 2 Background and Related Works

BNNs extend traditional neural networks by treating weights as random variables, allowing for the quantification of uncertainty in predictions. However, BNNs have reached far less popularity than their deterministic counterparts due to the high computational requirements and limited theoretical understanding. One of the major theoretical challenges concerns the comprehension of the parameter-space behavior of BNNs, in spite of the function one. Specifically, while it is established that under Gaussian initializations, the function distributions of wide BNNs converge to the Neural Network Gaussian Process (NNGP) limit (9; 10; 11; 12; 13; 14; 15), the dynamics of the posterior distribution remains less understood, with few exceptions (16; 17). We contribute to this literature by exploring sampling from the posterior distribution of wide BNNs, focusing on understanding its behavior and properties from a parameter-space perspective.

## 3 Proposed Method: pCN for Wide BNNs

Our approach leverages the pCN algorithm to sample from the posterior distribution of weights in wide BNNs. The key insight is to exploit the reparametrization proposed in (17) of the BNN weights that brings the posterior distribution closer to a standard Gaussian as the network width increases. Specifically, if we consider a BNN with L hidden layers and let $d^l$ be the width of the $l - th$ layer, the reparametrization is defined as:

$$\phi^{(l)} = \begin{cases} \Sigma^{-\frac{1}{2}}\theta^{(L+1)} - \mu & l = L+1 \\ \theta^{(l)} & else \end{cases} \tag{1}$$

where $\theta = [\theta^{(l)}]_{l=0,...,L} \sim \mathcal{N}(0, I_D)$ is the collection of flattened weights of the BNN and $\Sigma$ and $\mu$ are data-dependent terms defined as:

$$\Sigma = (I_{d^L} + \sigma^{-2}\Psi^T\Psi)^{-1} \quad \mu = \sigma^{-2}\Sigma\Psi^T y \tag{2}$$

Here, $\Psi$ represents the scaled output of the penultimate layer, and $\sigma^2$ is the observation variance. The convergence in the KL-divergence of the reparametrized posterior distribution to a standard Gaussian $\mathcal{N}(0, I_D)$ as the width goes to infinity suggests potential improvements in the mixing speed of MCMC procedures, compared to sampling from the notably arduous BNN posterior. Nevertheless, standard MCMC algorithms are notoriously ill-suited for the infinite-dimensional setting and must be carefully re-tuned as the dimension increases to avoid degeneracy in the acceptance probability (18; 19). To address these challenges, we focus on the robust pCN method, specifically designed to perform reliably in infinite-dimensional spaces.

The pCN algorithm (8) for sampling from the reparametrized posterior uses the following proposal:

$$\phi^* = \sqrt{1 - \beta^2}\phi + \beta w, \quad w \sim \mathcal{N}(0, I_D) \tag{3}$$

where $\beta \in [0, 1)$ is a stepsize parameter. The acceptance probability for this proposal is given by:

$$a(\phi^*|\phi) = min\{1, exp(-\ell(\phi) + \ell(\phi^*))\} \tag{4}$$

where $\ell$ is the log-likelihood.

Our main theoretical contribution is the following theorem:

**Theorem 3.1** *Consider the BNN model with the reparametrization 1. The acceptance probability of the pCN algorithm to sample from the reparametrized weight posterior, for any $\beta \in [0, 1)$, converges to 1 as the width of the network increases.*

This result has profound implications for sampling from wide BNNs. It guarantees that as the network becomes wider, the pCN algorithm becomes increasingly efficient, with nearly all proposals being accepted. This contrasts sharply with traditional MCMC methods, which often require careful tuning of step sizes to maintain reasonable acceptance rates in high dimensions. The proof Theorem 3.1 relies on the convergence of the empirical Neural Network Gaussian Process (NNGP) kernel to a constant independent of the weights as the network width increases and is reported in Appendix A.

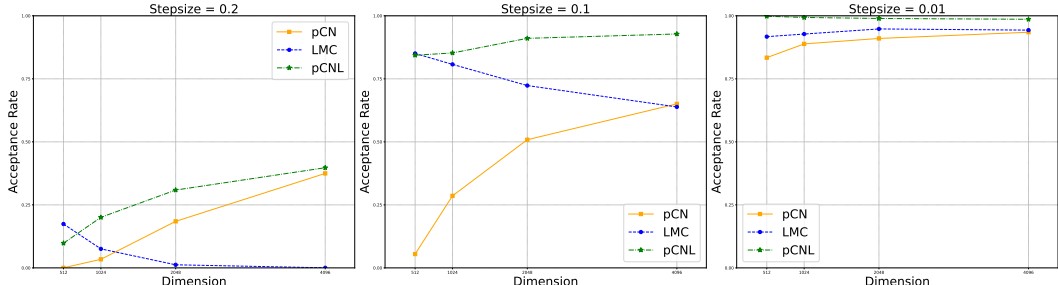

Figure 1: Comparison at different stepsizes ($\beta = 0.2, 0.1, 0.01$) of the acceptance probability obtained using: i. the underdamped LMC algorithm (or Metropolis Adjusted Langevin Algorithm: MALA); ii. the pCN algorithm; iii. the pCNL method. The neural network architecture used is a fully-connected with one hidden layer, and layer width that varies among the following values: $512, 1024, 2048, 4096$. The CIFAR-10 dataset is used, with the sample size fixed at $n = 256$. The acceptance rate of the pCN increases steadily as the width of the BNN grows at every stepsize, suggesting improved performance in wide BNNs and empirically confirming our theoretical analysis. The pCNL algorithm shows a similar trend in its acceptance rate, outperforming the other samples. In contrast, the LMC initially shows generally a deterioration in its acceptance rate as the width of the BNN increases, reflecting the sampler's non-robustness in high-dimensional settings.

## 4 Empirical Results

To validate our theoretical findings and demonstrate the effectiveness of the pCN sampler for wide BNNs, we conducted a series of experiments on the CIFAR-10 dataset. We replicate the setting in (17) using a fully-connected neural network with one hidden layer and varying the width from 512 to 4096 neurons. The code is available at github.com/lucia-pezzetti/Function-Space-MCMC-for-Wide-BNNs.

### 4.1 Acceptance Rate Convergence

Figure 1 shows the acceptance rates for pCN, underdamped Langevin Monte Carlo (LMC), and preconditioned Crank-Nicolson Langevin (pCNL) (8) samplers as a function of network width. The results clearly demonstrate that the pCN acceptance rate steadily increases as the network width grows, approaching 1 for very wide networks. This empirically confirms our theoretical prediction in Theorem 3.1. In contrast, the LMC sampler shows a decline in acceptance rate for wider networks, highlighting the advantage of pCN in high-dimensional settings. The empirical results also showcase that the pCNL, leveraging both the dimensional-robustness of the pCN and the gradient-informed proposal of the LMC, reaches the most desirable performances. This suggests the possibility to adapt Theorem 3.1 also to the Langevin version of the pCN sampler.

### 4.2 Effective Sample Size Analysis

Figure 2 presents the per-step Effective Sample Size (ESS) (20) for different samplers across various network widths. Both the pCN and pCNL samplers show a consistent increase in ESS as the network width grows, indicating improved sampling efficiency. This is particularly evident for larger step sizes ($\beta = 0.2$ and $\beta = 0.1$), where they outperform the LMC for wider networks. The increasing ESS demonstrates that the samplers not only maintains a high acceptance rate but also produces less correlated samples in high-dimensional spaces. The results for the smallest stepsize confirm the necessity of avoiding degeneracy in the stepsize, as this introduces autocorrelation among the collected samples and leads to a deterioration in their quality.

## 5 Discussion and Conclusion

In this paper, we investigated the effectiveness of the pCN sampler in sampling the posterior distribution of wide BNNs. Our method leverages recent theoretical insights into the behavior of wide neural networks and addresses the challenges of sampling in high-dimensional spaces. The key contributions

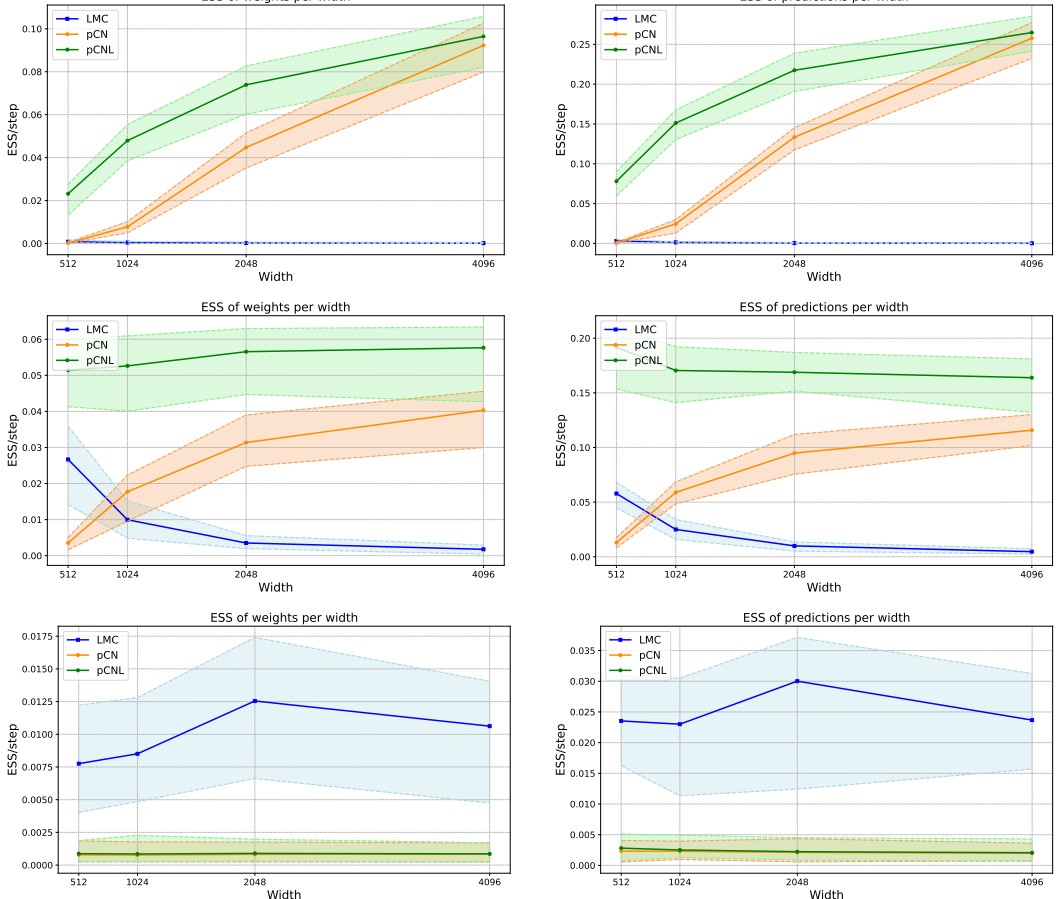

Figure 2: ESS analysis of the LMC, pCN and pCNL algorithms as a function of the 1-layer FCN's width for stepsizes $\beta = 0.2$ (above), $\beta = 0.1$ (middle) and $\beta = 0.01$ (below). The solid lines represent the average per-step ESS, whereas the shaded areas indicate the variability of the per-step ESS delineated by its minimum and maximum values. The setting used in the experiments is the same as the setting of Figure 1: the layer width of the BNN varies among the following values: $\{128, 512, 1024, 2096, 4192\}$. The CIFAR-10 dataset is used, with sample size fixed at $n = 256$. The poor LMC performance reflects the fact that standard MCMC procedures are ill-posed in high-dimensional settings. In contrast, the pCN and pCNL samplers demonstrate constant growth in ESS as the network width increases, indicating that enhancements in acceptance rate contribute positively to efficiency and performance. Finally, the smallest stepsize, $\beta = 0.01$, heavily affects the behavior of both algorithms, introducing high autocorrelation among the samples and affecting their quality.

of our work include theoretical guarantees for the convergence of the pCN acceptance probability to 1 as network width increases, and their empirical validation through extensive experiments on real-world datasets.

Our findings have significant implications for Bayesian deep learning, offering a scalable and robust approach to uncertainty quantification in large neural networks and contributing to the broader goal of combining the expressiveness of deep learning with the rigorous uncertainty quantification of Bayesian methods. Future directions for this work include extending the approach to more complex network architectures, such as convolutional and recurrent neural networks, and investigating theoretical guarantees for the pCNL sampler.

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

# A  Proof of Theorem 2.1

Let the assumptions of Theorem 3.1 hold. We start by analysing the general expression of the MCMC acceptance probability:

$$a = \min\left\{1, \frac{p(\phi^*|\mathcal{D})q(\phi|\phi^*)}{p(\phi|\mathcal{D})q(\phi^*|\phi)}\right\}.$$

We have already shown that $q(\phi^*|\phi) = \mathcal{N}(\sqrt{1-\beta^2}\phi, \beta^2 I_D)$. Regarding the reparamatrized weight posterior of the network, we observe that (17)

$$p(\phi|\mathcal{D}) = p(\phi^{(L+1)}|\phi^{(\leq L)}, \mathcal{D})p(\phi^{(\leq L)}|\mathcal{D})$$

$$\propto p(\phi^{(L+1)}|\phi^{(\leq L)}, \mathcal{D})\sqrt{det(\Sigma)}exp\left(\frac{1}{2}y^T(\sigma^2 I_n + \Psi\Psi^T)^{-1}y\right)$$

where $p(\phi^{(L+1)}|\phi^{(\leq L)}, \mathcal{D}) \sim \mathcal{N}(0, I_{d^{(L)}})$ is assured by the reparametrisation. It is then crucial to recognize the empirical NNGP kernel $\hat{K}_{\sigma^2} = \sigma^2 I_n + \Psi\Psi^T$ (21) and observe that $det(\Sigma) \propto det(\hat{K}_{\sigma^2})$.

Inserting everything in the expression of the acceptance probability we have:

$$\frac{p(\phi^*|\mathcal{D})q(\phi|\phi^*)}{p(\phi|\mathcal{D})q(\phi^*|\phi)} = \frac{p(\phi^{*(\leq L)}|\mathcal{D})p(\phi^{*(L+1)}|\phi^{*(\leq L)}, \mathcal{D})q(\phi|\phi^*)}{p(\phi^{(\leq L)}|\mathcal{D})p(\phi^{(L+1)}|\phi^{(\leq L)}, \mathcal{D})q(\phi^*|\phi)}$$

$$= \frac{p(\phi^{*(\leq L)})\sqrt{det(\Sigma^*)}exp\left(\frac{1}{2}y^T(\sigma^2 I_n + \Psi^*\Psi^{*T})^{-1}y\right)p(\phi^{*(L+1)}|\phi^{*(\leq L)}, \mathcal{D})q(\phi|\phi^*)}{p(\phi^{(\leq L)})\sqrt{det(\Sigma)}exp\left(\frac{1}{2}y^T(\sigma^2 I_n + \Psi\Psi^T)^{-1}y\right)p(\phi^{(L+1)}|\phi^{(\leq L)}, \mathcal{D})q(\phi^*|\phi)}$$

Where we denote with $\Sigma^*$ and $\Psi^*$ the covariance matrix and scaled input matrix of the redout layer in equation 2, but for a network with weights $\phi^*$ . Now:

$$q(\phi|\phi^*) \propto exp\left(-\frac{1}{2\beta^2}||\phi - \sqrt{1-\beta^2}\phi^*||^2\right)$$

$$= exp\left(-\frac{1}{2\beta^2}||\phi||^2 - \frac{(1-\beta^2)}{2\beta^2}||\phi^*||^2 + \frac{\sqrt{1-\beta^2}}{\beta^2}\phi^T\phi^*\right)$$

$$= exp\left(-\frac{1}{2\beta^2}||\phi||^2 - \frac{1}{2\beta^2}||\phi^*||^2 + \frac{1}{2}||\phi^*||^2 + \frac{\sqrt{1-\beta^2}}{\beta^2}\phi^T\phi^*\right)$$

From which

$$\frac{q(\phi|\phi^*)}{q(\phi^*|\phi)} = exp\left(\frac{1}{2}||\phi^*||^2 - \frac{1}{2}||\phi||^2\right)$$

and since

$$p(\phi^{*(\leq L)})p(\phi^{*(L+1)}|\phi^{*(\leq L)}, \mathcal{D}) \propto exp\left(-\frac{1}{2}||\phi^{*(\leq L)}||^2\right)exp\left(-\frac{1}{2}||\phi^{*(L+1)}||^2\right)$$

$$= exp\left(-\frac{1}{2}||\phi^*||^2\right)$$

we obtain

$$\frac{p(\phi^*|\mathcal{D})q(\phi|\phi^*)}{p(\phi|\mathcal{D})q(\phi^*|\phi)} = \frac{exp\left(-\frac{1}{2}||\phi^*||^2\right)exp\left(\frac{1}{2}||\phi^*||^2\right)\sqrt{det(\Sigma^*)}exp\left(\frac{1}{2}y^T(\sigma^2 I_n + \Psi^*\Psi^{*T})^{-1}y\right)}{exp\left(-\frac{1}{2}||\phi||^2\right)exp\left(\frac{1}{2}||\phi||^2\right)\sqrt{det(\Sigma)}exp\left(\frac{1}{2}y^T(\sigma^2 I_n + \Psi\Psi^T)^{-1}y\right)}$$

$$= \frac{\sqrt{det(\Sigma^*)}exp\left(\frac{1}{2}y^T(\sigma^2 I_n + \Psi^*\Psi^{*T})^{-1}y\right)}{\sqrt{det(\Sigma)}exp\left(\frac{1}{2}y^T(\sigma^2 I_n + \Psi\Psi^T)^{-1}y\right)}$$

$$\propto \frac{\sqrt{det(\hat{K}_{\sigma^2}^*)}exp\left(\frac{1}{2}y^T(\hat{K}_{\sigma^2}^*)^{-1}y\right)}{\sqrt{det(\hat{K}_{\sigma^2})}exp\left(\frac{1}{2}y^T(\hat{K}_{\sigma^2})^{-1}y\right)}$$

To conclude, we exploit the known convergence of the empirical NNGP kernel to a constant independent of $\phi^{\leq L} = \theta^{\leq L}$

$$\hat{K}_{\sigma^2} \to K_{\sigma^2} \quad \text{as } d_{min} \to \infty$$

This proves that the numerator and the denominator converge to the same quantity and, consequently, that their ratio converges to 1.

Implying the thesis

$$a = 1 \wedge \frac{p(\phi^*|\mathcal{D})q(\phi|\phi^*)}{p(\phi|\mathcal{D})q(\phi^*|\phi)} \to 1 \wedge 1 = 1 \quad \text{for } d_{min} \to \infty$$

# B Marginal-Conditional Decomposition

An alternative approach to Theorem 3.1 involves marginalizing the weights of the network's final layer and perform the sampling procedure only on the weights of the network's inner layers. This idea is effective since it acknowledges that exact sampling can be performed from the posterior distribution of the reparametrized weights of the last layer, once the weights from all preceding layers are known, i.e.,

$$p(\phi|\mathcal{D}) = p(\phi^{(L+1)}|\phi^{(\leq L)}, \mathcal{D})p(\phi^{(\leq L)}|\mathcal{D})$$
$$= p(\phi^{(L+1)}|\theta^{(\leq L)}, \mathcal{D})p(\theta^{(\leq L)}|\mathcal{D})$$

where the last equality follows directly from the reparametrisation definition and from the fact that $p(\phi^{(L+1)}|\theta^{(\leq L)}, \mathcal{D}) \sim \mathcal{N}(0, I_{d^{(L)}})$ for any fixed value of $\theta^{(\leq L)}$. The idea is then to simply perform pCN sampling on the posterior distribution over the inner-layers weights $\pi(\theta^{\leq L}|\mathcal{D})$. Then, once the samples

$$\left[\theta_i^{(\leq L)}\right]_{i=1,\ldots,n}$$

have been collected, we draw, $\forall i$, $\phi_i^{(L+1)} \sim \mathcal{N}(0, I_{d^{(L)}})$, to obtain a sample of the full posterior distribution of the reparametrised weights. On the pCN algorithm, the next theorem is a counterpart of Theorem 3.1:

**Theorem B.1** *Consider the BNN model with the reparametrisation 1, and set $p(\theta|\mathcal{D}) = p(\phi^{(L+1)}|\theta^{(\leq L)}, \mathcal{D})p(\theta^{(\leq L)}|\mathcal{D})$ and $\theta^{(\leq L)} = W$. Then, the acceptance probability of the pCN algorithm, for any $\beta \in [0, 1)$, applied to $p(\theta^{(\leq L)}|\mathcal{D})$ converges to 1 as the width of the network increases. If $d_{min}$ is the smallest among the network's layer widths, then*

$$a(\phi^*|\phi) = \min\left\{1, \frac{p(W^*|\mathcal{D})q(W|W^*)}{p(\phi|\mathcal{D})q(W^*|W)}\right\} \to 1 \quad as \ d_{min} \to \infty$$

## B.1 Proof of Theorem B.1

Since we are sampling only from the inner-weights $\theta^{(\leq L)} = W$ of the BNN, the acceptance probability becomes

$$a = 1 \wedge \frac{p(W|\mathcal{D})q(W|W^*)}{p(W|\mathcal{D})q(W^*|W)}$$

By definition 1, $p(W|\mathcal{D}) = p(\phi^{(\leq L)}|\mathcal{D})$ and thus we can exploit the known expressions from Appendix A:

$$p(W|\mathcal{D}) = p(W)\sqrt{det(\Sigma)}exp\left(\frac{1}{2}y^T(\sigma^2 I_n + \Psi\Psi^T)^{-1}y\right)$$
$$\propto exp\left(-\frac{1}{2}||W||^2\right)\sqrt{det(\Sigma)}exp\left(\frac{1}{2}y^T(\sigma^2 I_n + \Psi\Psi^T)^{-1}y\right)$$

Moreover

$$q(W|W^*) \propto exp\left(-\frac{1}{2\beta^2}||W - \sqrt{1-\beta^2}W^*||^2\right)$$

$$\implies \frac{q(W|W^*)}{q(W^*|W)} = exp\left(\frac{1}{2}||W^*||^2 - \frac{1}{2}||W||^2\right)$$

Putting all together and simplifying we get:

$$a = 1 \wedge \frac{\pi(W^*|\mathcal{D})q(W|W^*)}{\pi(W|\mathcal{D})q(W^*|W)}$$

$$= 1 \wedge \frac{\sqrt{det(\Sigma^*)}exp\left(\frac{1}{2}y^T(\sigma^2 I_n + \Psi^*\Psi^{*T})^{-1}y\right)}{\sqrt{det(\Sigma)}exp\left(\frac{1}{2}y^T(\sigma^2 I_n + \Psi\Psi^T)^{-1}y\right)}$$

That corresponds to the same exact expression found in Appendix A and hence convergence to 1 as the layers width increases is granted by the same arguments.

