# OpenReview forum: "Preconditioned Crank-Nicolson Algorithms for Wide Bayesian Neural Networks"
_NeurIPS.cc/2024/Workshop/BDU — NeurIPS BDU Workshop 2024 Poster_

### Official Review · Reviewer_iCTN · 2024-09-26
**Preconditioned Crank-Nicolson Algorithms for Wide Bayesian Neural Networks**

**Rating:** 7
**Confidence:** 1

**Review:**

This paper describes a new sampling algorithm for Bayesian Neural Networks. The key insight is that using a new reparameterization with the Crank-Nicolson sampling algorithm can lead to better acceptance rates in high dimensions. The paper is well-written and easy to follow, although most of the complexity is in the appendix. The theoretical and experimental data presented are convincing enough to warrant further investigation. My main criticism, which also reflects the early nature of the work, is that much more experimental work is needed to consider this sampling algorithm as a viable alternative, such as:

1) Comparison with more sophisticated sampling algorithms
2) Scaling up the experimental section with deeper (more than 1 layer) networks and more challenging tasks (including real-world problems)
3) Evaluating the downstream impact of the new parameterization, sampling, and higher acceptance rate on the quality of the posterior distribution, especially in high dimensions

However, I think this paper presents early evidence that this is an interesting direction to consider.

---

### Official Review · Reviewer_cKNp · 2024-10-02
**Interesting theoretical and empirical findings regarding pCN MCMC algorithms for wide BNNs**

**Rating:** 7
**Confidence:** 3

**Review:**

This paper presents interesting theoretical findings for preconditioned Crank-Nicholson algorithms for training wide Bayesian Neural Networks. They find that the acceptance rate and efficiency of this algorithms improves with the width of the BNNs. They tested this insight on numerical experiments on CIFAR-10.

---

### Decision · Program_Chairs · 2024-10-09

Accept (Poster)